# Effects of Oral Nutritional Supplementation on Patients with Venous Ulcers: A Clinical Trial

**DOI:** 10.3390/jcm11195683

**Published:** 2022-09-26

**Authors:** Paulla Guimarães Melo, João Felipe Mota, Cynthia Assis de Barros Nunes, Suelen Gomes Malaquias, Alexandre Siqueira Guedes Coelho, José Verdú Soriano, Maria Márcia Bachion

**Affiliations:** 1Graduate Program in Health Sciences, Federal University of Goias, Goiânia 74605-050, Brazil; 2School of Nutrition, Federal University of Goias, Goiânia 74605-050, Brazil; 3National Council for Scientific and Technological Development—CNPq, Brasilia 71605-001, Brazil; 4School of Nursing, Federal University of Goias, Goiânia 74605-080, Brazil; 5Graduate Program in Nutrition and Health, Federal University of Goias, Goiânia 74605-080, Brazil; 6School of Health Sciences, University of Alicante, 03690 San Vicente del Raspeig, Spain

**Keywords:** varicose ulcer, body composition, eating, oral nutritional supplementation

## Abstract

Background: The dosage and safety of nutritional supplements for patients with venous ulcers are still not well established. Aim: To evaluate the effects of a high calorie, high protein, arginine-, zinc-, and vitamins A, C, and E-enriched nutritional supplement on the biochemical profile, dietary intake, anthropometry, muscle strength, and characteristics of lesions of patients with venous ulcers. Methods: A controlled before–after clinical trial with a four-week follow-up involved 27 patients with venous ulcers under outpatient treatment in Brazil. It was administered in two to three doses per day (200 mL each) of a high-calorie and high-protein supplement enriched with arginine, zinc, and vitamins A, C, and E. Patients were assessed for anthropometric parameters, dietary intake, biochemical tests, and healing conditions according to the Pressure Ulcer Scale for Healing (PUSH). Results: It was observed that an increase in energy and protein supply led to an adequate intake of immunonutrients (zinc and vitamins A, C, and E), increased body weight, increased body mass index, and stronger handgrip strength. The injury area and the score on the PUSH notably decreased after the intervention (*p* < 0.001). Conclusions: The administered supplement, at the tested dosage, improved the nutritional status and characteristics of lesions in patients with venous ulcers.

## 1. Introduction

Venous leg ulcers (VLU) are skin injuries that result from chronic venous insufficiency, and they require a long healing time [1]. It is known that it takes up to 12 months for 93% of the injuries to epithelize, and that 7% do not heal even after five years of adequate treatment [2]. Several factors can influence the healing process, and one of them is nutritional status [3,4].

Healing of chronic ulcers involves biochemical, molecular, and cellular processes, which need nutritional substrates to promote cell proliferation, enzymatic activity, and protein synthesis [3,4]. The amino acid arginine [5,6]; omega-3 fatty acids (eicosapentaenoic and docosahexaenoic acids) [7,8]; vitamins A, C, and E; and minerals such as copper, iron, and zinc [9] stand out among those nutritional substrates. The absence of these immunonutrients impairs healing, since it triggers a weakening of the immune system [3,4,8].

Patients with chronic leg ulcers show diminished levels of vitamin C [10,11,12], vitamin A, vitamin E [13], vitamin B6, vitamin B9 [12], omega-3, zinc [11,12,13], vitamin D [12,14], and protein [12]. Although this may be a typical characteristic of the industrialized population, affecting even elderly people [15], it is necessary to consider that the presence of this type of injury, and healing itself, increases the demand for nutrients, which adds to insufficiencies related to inadequate intake [3,4,16]. Unsatisfactory nutritional status can impact the healing prognosis of people with venous ulcers [17]. Consequently, it is important to promote better nutrient supply conditions, which may result in indications for nutritional supplementation.

Nutritional recommendations are well established for healthy people [18,19], patients with pressure injuries (PI) [9,20,21], and patients with diabetic foot ulcer [22,23,24]. However, the guidelines for people with other types of chronic injuries, such as VLU, have not yet been clearly defined, because studies in this area are rare [4,25,26].

Oral nutritional supplements are products made up of specific or combined nutrients, and can be presented in liquid form, ready for consumption, or in powder form, requiring, in this case, dilution in beverages or food [27]. They are used to achieve better clinical outcomes and indicate when the individual needs a greater nutritional intake, which is not being achieved only by food consumption, and has a functioning and accessible gastrointestinal tract [28].

There are no recommendations regarding composition, dose, and time of use of supplements for this population [4,20,21,25]. So far, only one other study has investigated the use of oral nutritional supplementation in this population [29], but the authors did not assess anthropometric parameters and food intake. Different formulations and doses have been studied in people with chronic ulcers, from specific nutrient modules [14,30,31] to commercially available oral nutritional supplements that show complete nutritional composition, including calorie and protein supply and nutrients that specifically enhance healing [32,33,34,35].

Randomized clinical trials with patients with PI have shown that high calorie, high protein, arginine-, zinc-, and antioxidant vitamin (vitamins A, C, and E)-enriched nutritional supplements (Cubitan^®^, Danone Nutricia) improved immune function and protein supply, and promoted better healing rates [33,34,35]. These benefits seem to have been present even in people who did not have malnutrition [34]. Supplementation recommendations must take into account overall diet quality, and propose a nutrient profile that favors healing by considering all the aspects related to nutritional deficiency, even in people who are overweight or obese [25].

To better understand the effects of nutritional supplementation on healing in people with VLU, it is fundamental to assess nutritional gains in this population, including changes in body weight, stamina, and muscle strength, but a gap remains in this research field [25]. Additionally, since this population mainly receives care in primary health care settings, it is paramount to focus on a supplement that is effective, safe, easy to use, and can be incorporated into the health system in outpatient care protocols.

The objective of the present study is to evaluate the effects of a high-calorie and high-protein supplement enriched with arginine, zinc, and vitamins A, C, and E on the biochemical profile, dietary intake, anthropometric markers, and muscle strength of patients with venous ulcers, as well as the characteristics of their injuries, over a four-week outpatient treatment period.

## 2. Materials and Methods

### 2.1. Study Design and Participants

This was a controlled before–after clinical trial with a four-week follow-up involving 27 patients with VLU under outpatient treatment in Goiânia, a state capital located in the Center-West region of Brazil, who were referred to the research unit or sought it spontaneously. The study design can be seen in Figure 1. The eligibility criteria were participants over 18 years old or older with active VLU, classified as C6 on the Clinical, Etiological, Anatomical, and Pathophysiological classification (CEAP C6) [36], and an ankle brachial index ranging from 0.8 to 1.3, including these limit values [37]. Patients with severe hearing or visual loss, neoplasia diagnosis, kidney or liver disease, or a Mini-Mental State Examination score lower than that recommended for their level of education [38,39,40] were excluded from the study.

### 2.2. Intervention Protocol, Follow-Up, and Adverse Events

The intervention was administering a high-calorie, high-protein oral nutritional supplement that was enriched with specific nutrients for healing, such as arginine, zinc, and vitamins A, C, and E (Cubitan^®^, Danone Nutricia, 200 mL). This supplement was chosen because it is protein-rich, contains immunonutrients that play a role in the healing process, has shown positive results in clinical trials in people with chronic injuries [32,33,34,35], and is included in the set of supplies provided by the Goiânia Municipal Health Secretariat to patients with PI. The dosage was defined based on injury area. Patients with an ulcer smaller than 20 cm^2^ received two bottles a day, and those whose ulcer had an area of at least 20 cm^2^ got three bottles a day over four weeks. This dosage was proposed according to the protocols described in other clinical trials for the analysis of the treatment of patients with chronic ulcers [33,34]. All the patients were instructed to maintain their usual diet, and received treatment for their injuries based on the recommendations of best practices in VLU management [1,2,4,41], including compression therapy.

The therapy used in the local service consisted of daily dressings, performed by the nursing staff, as follows: cleansing with warmed saline solution; followed by compression with 2% polyhexamethylene biguanide solution for 15 min; covering with 2% papain gel or hydrogel, rayon gauze, or cotton gauze for absorption of exudate; bandaging the metatarsal region up to four centimeters before the patellar region with a crepe bandage; and applying short-stretch and single-layer compression bandages in the same region.

The patients received daily doses of the nutritional supplement, the ingestion of which was supervised on weekdays at the clinic where the study was carried out. The consumption of the supplement on weekends occurred in the patients’ homes. The packages were given back to the researchers, and the patients recorded their daily nutritional supplement intake on an individual form that was verified during every follow-up appointment. The adopted criterion of adherence to taking the supplement was consuming at least 80% of the recommended weekly dosage [42]. The patients who did not reach this minimum level of consumption were excluded from the study. The presence of possible adverse effects associated with the use of the supplement, such as nausea, decrease in appetite, or gastric discomfort, was investigated during every daily follow-up appointment. If these effects manifested with mild to moderate intensity, it was seen as an indication for more fractionated consumption, and the result of this management was evaluated afterwards.

### 2.3. Socioeconomic Characteristics and Associated Comorbidities

Socioeconomic data were collected by applying a questionnaire. Associated comorbidities were reported by the participants and confirmed by consulting prescriptions of drugs for continuous treatment. Perceived stamina after use of the supplement was evaluated by asking the following question: “Did you notice an improvement in your stamina to carry out daily activities over the past four weeks?” The answer options were “Yes” and “No.”

### 2.4. Dietary Intake and Anthropometric Measures

Dietary intake was assessed before and after supplementation. Three 24 h food diaries were applied, two relating to consumption during weekdays and one pertaining to diet on a weekend day. Food quantities, reported in common household measures by the participants, were converted into grams or milliliters. All the data in the food diaries were treated by using Dietpro^®^ software version 5.8 (Viçosa, Brazil).

Body weight and height were measured by using a portable electronic scale and a mobile stadiometer, respectively, as recommended by Lohman, Roche, and Martorell [43]. Body mass index (BMI) was calculated by dividing weight by height squared, expressed in kg/m^2^, and categorized according to the guidelines suggested by the World Health Organization [44] when the patient was an adult and by Lipschitz [45] in the case of elderly people. Arm circumference (AC), arm muscle circumference (AMC), and triceps skin-fold thickness (TSF) were measured at the middle upper arm. Circumferences were determined by using an inelastic measuring tape, in accordance with the recommendations by Lohman, Roche, and Martorell [43]. The final TSF value was the average of three measurements, using a Lange Skinfold^®^ caliper. The following equation was applied: AMC (cm) = AC (cm) − [TSF (mm) × 0.314] [43].

Handgrip strength (HS) was obtained by selecting the best result among three consecutive measurements five seconds apart. A Crown^®^ portable mechanical dynamometer was used, following the instructions described by Watanabe et al. [46].

### 2.5. Injury Area, Healing Conditions, and Biochemical Data

Planigraphic images of the injuries were obtained during the initial and final evaluations, and their areas were calculated by using Image J^®^ software version 1.49 (Laboratory for Optical and Computational Instrumentation, University of Wisconsin, Madison, WI, USA, 2016), so the supplement dosage could be established and healing could be better assessed. If the patient had more than one injury, the considered area was the sum of the areas of all the injuries.

The Pressure Ulcer Scale for Healing (PUSH) was applied to evaluate healing conditions [47,48]. This instrument includes the following parameters: injury area, exudate quantity, and worst tissue on the wound bed. Although this scale was initially designed to assess PI, it began to be used to evaluate healing in VLU [49,50]. The total score could range from 0 to 17, with the highest score being considered the worst healing condition [48].

Blood samples were collected one week before the beginning and one week after the end of the intervention. Fluorescence flow cytometry and microscopy were used to determine hematocrit and hemoglobin concentration, respectively. Immunoturbidimetry was used to measure C-reactive protein concentration, and the enzymatic colorimetric method was used for the determination of fasting glycemia and albumin concentrations.

### 2.6. Statistical Analyses

A database was created by using Microsoft Excel^®^ software version 2010 (Microsoft Office, Microsoft Corporation, Redmond, WA, USA). First, the variables were treated by applying descriptive statistics, with the determination of frequency distribution and measures of central tendency (average and median) and dispersion (standard deviation). Perceived stamina was assessed by using Fisher’s exact test. Adherence of the quantitative variables to a normal distribution was evaluated by using the Shapiro–Wilk test. Values obtained before and after supplementation were compared by applying the paired Student’s *t*-test or the Wilcoxon signed-rank test, depending on the distribution of the variables. The critical level of significance adopted in the tests was 0.05. Statistical analyses were run on R software version 3.4.4 (R Core Team, R Foundation for Statistical Computing, Vienna, Austria, 2018).

## 3. Results

### 3.1. Baseline Characteristics

Initially, 29 patients were included in the sample, but two were withdrawn from the follow-up for personal reasons with no connection to the intervention. The majority of the sample were women (63%, *n* = 17), more than half were 60 years old or older (55.6%, *n* = 15), and they had low average per capita income and low level of education (Table 1). Nearly one-third (29.6%, *n* = 8) had a diagnosis of diabetes mellitus, and more than half (51.8%, *n* = 14) had hypertension. Alcoholism affected 11.1% (*n* = 3) of the sample, and 3.7% (*n* = 1) were smokers.

The initial characteristics of VLU, injury duration, injury area, and PUSH score varied considerably (Table 1). The injuries ranged from small to extensive, lasted between months and years, and showed various healing conditions.

### 3.2. Dietary Intake

Most participants (74.1%) had a normocaloric diet (median = 24.51 kcal/kg/day) before supplementation and, therefore, their diet did not meet their calorie needs. In addition, 77.8% of the patients had low consumption of protein (0.96 g/kg/day), and vitamin A (average = 146.2 ± 102.3 μg/day) and magnesium (median = 250.4 mg/day) stood out (Table 2 and Table 3).

The use of nutritional supplementation increased calorie intake (28.0 ± 12.1 to 32.1 ± 10.1 kcal/kg/day; *p* < 0.001), protein consumption (77.8 ± 33.2 vs. 122.1 ± 32.4 g/kg/day; *p* < 0.001), and, most markedly, arginine supply (3.9 ± 1.8 vs. 10.4 ± 1.8 g/day; *p* = 0.000) (Table 2). The carbohydrate intake percentage decreased (58.4 ± 8.3 vs. 52.0 ± 6.1 %; *p* < 0.001), whereas the lipid and protein intake percentages increased (27.0 ± 6.0 vs. 28.7 ± 4.2 %; *p* < 0.001 and 14.4 ± 3.2 vs. 18.9 ± 3.2 %; *p* < 0.001; respectively). The consumption of monounsaturated and polyunsaturated fatty acids increased with the addition of the supplement (Table 2). The consumption of all the analyzed micronutrients increased substantially, especially vitamin A (146.2 ± 102.3 vs. 710.3 ± 144.2 μg/day; *p* < 0.001) and selenium (90.4 ± 34.5 vs. 235.3 ± 36.2 μg/day; *p* < 0.001) (Table 2).

The administration of the nutritional supplement led to adequacy in the consumption of iron, copper, zinc, selenium, vitamin C, and protein, without exceeding the tolerable upper intake level (TUIL), based on the dietary reference intakes (Table 3). It is noteworthy that vitamin A levels did not reach normality, despite a considerable increase after supplementation, as described above (Table 2).

### 3.3. Anthropometric Measures, Follow-Up, and Adverse Effects

Body weight (83.2 ± 16.6 vs. 84.2 ± 16.3 kg, *p* = 0.003), BMI (31.3 ± 5.1 vs. 31.7 ± 4.9 kg/m^2^, *p* = 0.002) and handgrip strength (median = 21 vs. 24 kgf, *p* = 0.010) increased substantially after four weeks of use of supplementation (Table 4).

Mild adverse effects were reported by 22.2% of the participants (*n* = 6), nausea by 11.1% (*n* = 3), and nausea and decreased appetite by 11.1% (*n* = 3). Patients consumed at least 80% of the offered supplement. After the intervention, 77.8% of the participants (*n* = 21) stated that they had perceived an improvement in their stamina (*p* = 0.011).

### 3.4. Injury Area, Healing Conditions, and Biochemical Data

Comparison with the baseline data (Table 1) indicated significant reduction in both injury area (*p* < 0.001, Wilcoxon test) and PUSH scores (*p* < 0.001, Wilcoxon test). The nutritional supplement did not affect fasting glycemia, hemoglobin, hematocrit, albumin, or CRP concentrations (Table 4).

## 4. Discussion

As far as can be determined, this is the first clinical trial that evaluated the effects of a high-calorie, high-protein, and immunonutrient-enriched nutritional supplement on dietary intake, anthropometry, biochemical profile, and healing conditions in patients with VLU. Nutritional intervention studies in people with venous ulcers are rare [25,26] and have usually tested only one nutrient [14,51] or a set of nutrients from the same group, such as flavonoid-rich nutraceutical (a formulation with diosmin, hesperidin, rutin, astaxanthin, horse chestnut, and althea) [52]. Only one study tested the use of a multicomponent nutritional supplement; however, the sample consisted of only two participants, which implies a high risk of bias [53].

The administered oral nutritional supplement improved dietary intake and led to an increase in total energy intake and consumption of macronutrients, arginine, MUFA, PUFA, and all the analyzed micronutrients. It also increased body weight and HS, improved perceived stamina, and contributed to better healing conditions, such as a decrease in injury area and PUSH score.

Documents addressing PI have indicated that people with chronic injuries may show nutritional risk because of inadequate intake of nutrients that are important for healing by means of food, and that prescribing supplements can be an option in these cases [9,21]. Studies have shown that use of oral supplements enriched with protein, arginine, and micronutrients improved the nutritional state of the examined people [33,34,35,54].

The literature points out the existence of nutritional imbalance in people with venous ulcers [10,11,14,55]. Therefore, the results reported in the present study can be considered highly positive, since the supplement contributed to fixing the dietary deficits of the participants, which would be unlikely to be corrected with food consumption alone in a short period such as a four-week follow-up.

A study using the same nutritional supplement at a dosage of 200 mL three times a day over eight weeks, did not find a significant increase in serum zinc levels and higher concentrations of vitamin C (*p* = 0.015) [34]. Additionally, higher PUSH scores in the intervention group (*p* = 0.011) indicated better healing conditions, and there was an increase in body weight and BMI when the intervention was completed [34]. The findings of the present study corroborated these results, although its follow-up lasted half as long as that in the cited study.

A study in which the participants were elderly people with PI found an increase in serum zinc (*p* < 0.01) and in the dietary intake of protein (*p* < 0.001), arginine (*p* < 0.001), zinc (*p* < 0.001), and vitamin C (*p* < 0.001) after supplementation with Cubitan^®^ (Danone Nutricia) at a daily dosage of 400 mL over 12 weeks [33]. The authors also emphasized the association between use of the supplement and a significant improvement in PUSH scores (*p* < 0.05). The present study had a similar finding, but with a shorter intervention period.

The daily magnesium intake exceeded the TUIL recommended for healthy individuals (median = 430 mg after supplementation vs. TUIL = 350 mg) [18], which did not necessarily indicate an overdose for people with VLU. In contrast, the dietary reference intake of some nutrients [18], such as vitamin A, was not reached after the intervention. These results showed the need to create individualized diet plans that meet patients’ requirements, especially regarding vitamin A supply.

Nausea was also reported as an adverse effect in another study that analyzed the effects of the same oral nutritional supplement after an eight-week intervention, together with diarrhea, constipation, and dyspepsia [34], which were not mentioned by the participants in the present study.

In a prospective 12-week study investigating the effect of Cubitan^®^ (Danone Nutricia) oral nutritional supplementation in patients with VLU, the authors only reported that the oral therapy was well tolerated and did not mention any complaints from patients regarding the intake of the supplement [29]. The authors observed an increase in pre-albumin levels in the first six weeks of intervention, with stabilization during the sixth to eighth week of follow-up, but these data were not statistically significant [29]. Traditionally, albumin and pre-albumin values were biochemical tests considered useful for the assessment of patients’ nutritional status, with low levels indicating protein-energy malnutrition [56]. Currently, these tests are considered to be more closely related to the state of inflammation, which promotes in the body a hepatic reprioritization of protein synthesis and redistribution of serum proteins through increased capillary permeability. Thus, in the presence of a disease that promotes a state of inflammation, serum albumin and prealbumin should not be used as nutritional markers [56].

Handgrip strength is an anthropometric measure that stands out as simple and is commonly used in clinical practice, since it is noninvasive [57] and is related to total muscle strength. It has been considered a good indicator of nutritional status [57,58], and is increasingly used because of its ability to monitor post-supplementation improvements [48] and help identify people at nutritional risk [59].

The present study found a significant increase in HS in the final evaluation. Nevertheless, values below the 30th percentile were identified, a result that was lower than expected, given that the general population is between the 25th and 75th percentiles [60,61]. This increase probably occurred because muscle protein stocks quickly respond to a status of restored nutrition [57]. It is known that, in situations of muscle mass depletion, skeletal muscles become a preferential source of energy (since protein stocks are low), which results in lower muscle strength and functioning [58,59]. This finding is relevant for the population affected by VLU, since it will be possible to use nutritional supplementation to improve physical activity, a factor that increases blood flow in the lower limbs and consequently contributes to healing VLU.

The participants took daily 30 min walks six times a week, which may have led to better nutrient absorption and increased HS, TSF, and AMC, suggesting that the participants’ body weight gain was related to muscle mass gain. Nutritional supplementation is carried out over an established period as an adjuvant therapy in the healing process, and, regardless of its use, being overweight must be considered a risk factor for the development of VLU [25,62]. Consequently, it is recommended that individual diet plans be prepared so people with VLU have their nutritional needs met, and a healthy and sustainable diet is put into practice to allow patients to reach a normal body weight [25].

Liquid oral nutritional supplements lead to cost savings in the hospital setting, since they contribute to lowering mortality, complication rates, and lengths of stay [63]. These products can also offer functional benefits, such as mobility and ability to carry out self-care and usual daily activities [64].

The improvement in stamina associated with the use of the supplement may result in greater adherence to physical exercises that facilitate blood flow in the legs and are considered adjuvants in the treatment of venous ulcers [65,66,67]. Results include minimization of venous hypertension, edema reduction, and pain relief [67,68].

One of the limitations of the present study was inherent in its design, that is, the absence of a parallel control group, which precluded comparisons between groups to verify the effects of the nutritional supplement alone. The short observation time can be considered a limitation, but the option of carrying out the intervention for four weeks seemed more careful because at the time of the research there was no information available on the safety of prolonged use of the supplement for this population. Although the intervention showed significant results on wound healing, a follow-up for 12 weeks could show better benefits for the injury healing process.

Another limitation was the reduced sample size. However, this characteristic is often present in studies that have analyzed patients with VLU [25]. It is also important to emphasize that methods considered to be the gold standard for body composition analysis, such as bioimpedance and dual-energy X-ray absorptiometry, were not applied, since the present study aims to show that it is possible to carry out this analysis by using simple methods that are part of clinical practice, and can be incorporated into future protocols for care of people with VLU.

## 5. Conclusions

High calorie, high protein, immunonutrient-enriched oral nutritional supplementation improved immunonutrient intake and stamina and contributed to increasing body weight, muscle mass, and HS and decreasing PUSH scores and injury area in patients with venous ulcers. The results indicated that the dosage was safe and sufficient to improve most of the analyzed parameters. The findings allow the authors to recommend oral nutritional supplementation for patients with venous ulcers in the initial stage of the treatment, that is, the first four weeks, as an adjuvant to the prescribed therapy.

## Figures and Tables

**Figure 1 jcm-11-05683-f001:**
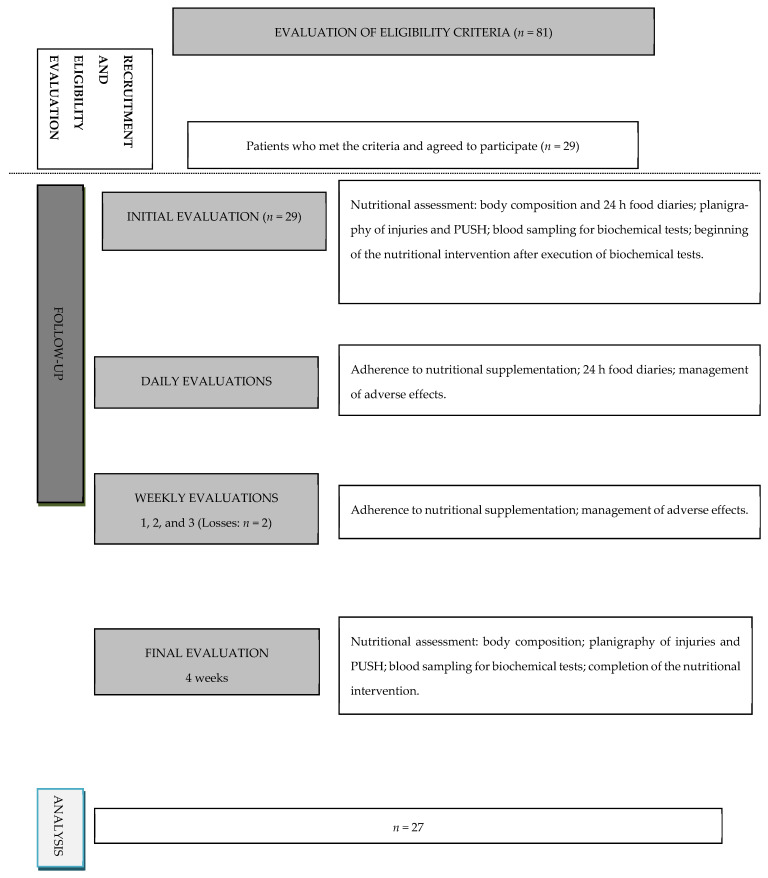
Flowchart showing care and follow-up of the patients analyzed in the study.

**Table 1 jcm-11-05683-t001:** Baseline characteristics of the patients and their venous ulcers.

Variable	Average	SD	Median	Minimum	Maximum
Age (years)	59.4	11.2	61.0	42.0	86.0
Per capita income (US$ *)	235.4	172.2	191.0	57.31	859.6
Level of education (years)	6.3	4.7	6.0	0.0	13.0
Injury duration (months)	33.3	48.0	6.0	1.0	192.0
Injury area (cm^2^)	11.3	12.1	5.9	0.5	40.3
PUSH score	11.7	3.3	12.0	5.0	17.0

* USD exchange rate (US$) in 2016: US$1 = R$3.49; SD = standard deviation; PUSH = Pressure Ulcer Scale for Healing.

**Table 2 jcm-11-05683-t002:** Dietary intake of patients with venous ulcers before and after nutritional supplementation.

Variable	Initial	Final	*p*-Value ^a,b^
TEI (kcal/day) *	2023.9 (1627.9–2608.2)	2493.9 (2073.5–2989.8)	<0.001 ^b^
TEI (kcal/kg/day) *	24.5 (17.9–36.9)	29.3 (26.5–7.7)	<0.001 ^b^
CHO (g/day)	343.1 ± 179.4	344.8 ± 116.5	0.895 ^a^
CHO (%/day)	58.4 ± 8.3	52.0 ± 6.1	<0.001 ^a^
CHO (g/kg/day) *	3.6 (2.5–6.1)	3.9 (3.2–5.2)	0.691 ^b^
Simple carbohydrates (g/day)	130.1 ± 93.5	105.9 ± 53.3	<0.001 ^a^
LIP (g/day)	63.6 ± 22.1	84.4 ± 29.5	0.140 ^a^
LIP (%/day)	27.0 ± 6.0	28.7 ± 4.2	<0.001 ^a^
LIP (g/kg/day)	0.79 ± 0.3	1.0 ± 0.3	<0.001 ^a^
MUFA (g/day)	22.7 ± 7.0	33.6 ± 9.9	<0.001 ^a^
PUFA (g/day)	13.2 ± 5.9	19.0 ± 8.5	<0.001 ^a^
SAT (g/day)	21.1 ± 9.1	24.9 ± 10.3	0.085 ^a^
TRANS FATS (g/day)	0.6 ± 2.1	0.22 ± 0.24	<0.001 ^a^
PTN (g/day)	77.9 ± 33.2	122.1 ± 32.4	<0.001 ^a^
PTN (%/day) *	14.9 (11.9–16.4)	19.0 (16.4–20.8)	<0.001 ^b^
PTN (g/kg/day)	0.96 ± 0.40	1.5 ± 0.4	0.000 ^a^
Arginine (g/day) *	3.6 (2.6–4.7)	10.3 (9.5–11.6)	0.000 ^b^
Vitamin A (μg/day)	146.2 ± 102.3	710.3 ± 144.1	<0.001 ^a^
Vitamin C (mg/day)	242.2 ± 237.4	743.2 ± 193.0	<0.001 ^a^
Iron (mg/day)	12.0 ± 5.9	24.9 ± 5.2	<0.001 ^a^
Magnesium (mg/day) *	250.4 (190.3–318.8)	430.5 (360.6–494.6)	<0.001 ^b^
Copper (mg/day) *	1.1 (0.8–1.3)	2.7 (2.7–4.0)	<0.001 ^b^
Selenium (μg/day)	90.4 ± 34.5	235.3 ± 36.2	<0.001 ^a^
Zinc (mg/day) *	11.1 (8.1–14.2)	30.5 (27.2–35.1)	<0.001 ^b^

Data were presented in mean ± standard deviation or * median (1st and 3rd quartiles); ^a^
*p* value calculated by using Student’s *t*-test; ^b^
*p* value calculated by using the Wilcoxon test; TEI = total energy intake; CHO = carbohydrate; LIP = lipid; PTN = protein; MUFA = monounsaturated fatty acids; PUFA = polyunsaturated fatty acids; SAT = saturated fatty acids.

**Table 3 jcm-11-05683-t003:** Dietary intake of patients with venous ulcers before and after nutritional supplementation and comparison with dietary reference intakes *.

Nutrient	Before Supplementation (%)			After Supplementation (%)			*p*-Value
	Below	Normal	Above **	Below	Normal	Above **	
Iron	29.6	70.4	0	0	0	100	0.007
Magnesium	74.1	11.1	14.8	14.8	0	85.2	<0.001
Copper	33.3	66.7	0	0	0	100	<0.001
Selenium	18.5	81.5	0	0	0	100	0.062
Zinc	29.6	70.4	0	0	0	100	0.001
Vitamin A	100	0	0	67.7	0	32.3	0.003
Vitamin C	26.9	73.1	0	0	0	100	0.015
TEI (kcal/kg/day)	74.1	3.7	22.2	55.6	11.2	33.2	1
PTN (g/kg/day)	77.8	7.4	14.8	48.1	18.5	33.4	<0.001

* Dietary reference intakes for micronutrients—source: IOM, 2005; TRUMBO et al., 2002; TEI = total energy intake (reference value = 30 to 35 kcal/kg/day)—source: POSTHAUER et al., 2015. PTN = protein (reference value = 1.25 to 1.5 g/kg/day)—source: POSTHAUER et al., 2015. ** The nutrient levels are higher than dietary reference intakes (recommended daily intake by age group and sex), but did not exceed the tolerable upper intake level. *p* values were calculated by using the Wilcoxon test.

**Table 4 jcm-11-05683-t004:** Biochemical and body composition variables of patients with venous ulcers before and after nutritional supplementation.

Variable	Initial	Final	*p*-Value ^a,b^
Hematocrit (%)	41.5 ± 5.0	41.6 ± 4.1	0.932 ^a^
Hemoglobin (g/dL)	13.6 ± 1.6	13.85 ± 1.36	0.156 ^a^
Fasting glycemia (mg/dL) *	97.0 (86.9–107.5)	96.0 (84–105)	0.779 ^b^
Albumin (g/dL) *	4.0 (3.6–4.3)	4.1 (3.7–4.3)	0.794 ^b^
CRP (mg/L) *	6.0 (6–6)	6.0 (6–8.5)	0.425 ^b^
Body weight (kg) *	88.0 (71.4–94.7)	88.0 (72.5–94.9)	0.003 ^b^
BMI (kg/m^2^) *	31.3 (28.3–34.4)	31.34 (28.7–40.7)	0.002 ^b^
AC (cm)	33.7 ± 4.4	33.9 ± 4.2	0.207 ^a^
AMC (cm)	25.9 ± 2.8	26.3 ± 3.1	0.179 ^a^
TSF (mm)	24.7 ± 10.3	24.2 ± 9.7	0.532 ^a^
HS (kgf) *	21.0 (17.5–33)	24.0 (18–34)	0.010 ^b^

Data were presented in mean ± standard deviation or * median (1st and 3rd quartiles); ^a^
*p* value calculated by using Student’s *t*-test; ^b^
*p* value calculated by using the Wilcoxon test; CRP = C-reactive protein; BMI = body mass index; AC = arm circumference; AMC = arm muscle circumference; TSF = triceps skin-fold thickness; HS = handgrip strength.

## Data Availability

The data used to support the findings in this study are available from the corresponding author upon reasonable request.

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
