# Peer review of "Effects of Oral Nutritional Supplementation on Patients with Venous Ulcers: A Clinical Trial"

_jcm, 2022, doi:10.3390/jcm11195683_

Round 1

Reviewer 1 Report

I am glad that the authors emphasized that there is a gap in the nutritional guidelines that are well developed for healthy people and with certain diseases, and that there is a lack of well-designed guidelines for patients with various types of chronic wounds (except for pressure ulcers). For this reason, studies such as this could contribute to the advancement of research into the specific nutritional needs of chronic wound patients.
I have a few remarks and comments about this work:

1. Please expand the description of the methods of treatment and care of VLUs, (What local treatment strategy was used, what criteria of dressing selection; compression therapy - in what form: single or multi-layer bandages, short- or long-stretch, compression stockings?) What compression class?

2. I believe that the limitation of the study is the short observation time - 4 weeks. In the case of VLUs, the optimal follow-up time is 12 weeks. While, from the point of view of the ONS itself and its impact on blood parameters (including albumin levels), 4 weeks may be sufficient to supplement the deficiency. However, a follow-up after this time could show even better benefits for the injury healing process. In a similar study, Mościcka P et al. Showed that albumin levels increase the most until the 6th week of supplementation, then they are stable. In contrast, the healing of injuries was most effective between the 6-8 weeks of follow-up. It can be assumed that healing is accelerated after individual nutritional deficiencies are compensated: https://doi.org/10.5114/ada.2021.104730

3. In my opinion, the introduction to the work should include a definition and development of Oral Nutritional Supplementation (ONS). I also propose to change the keyword "Nutrition" to "Oral Nutritional Supplementation" for a more accurate selection of publications in databases.

4. I have not found references 5, 30 and 31 in the text.

Author Response

Authors’ answer 1 : We thank you for the suggestion and would like to inform you that a more detailed description of how the dressings were performed in the participants has been included in the second paragraph of the Materials and Methods section, item 2.2 (Intervention protocol, follow-up, and adverse events).

Authors’ answer 2: Thank you for raising this. We agree with the reviewer that a longer intervention time would bring better results. This was addressed in the discussion section when describing the limitations of the study. We are grateful for the suggestion of a recently published article in this area (Mościcka P et al., 2022). We have also included a citation about it in the introduction, mentioning that studies with this population are rare, in the sixth paragraph; and in the discussion section. 

Authors’ answer 3: We thank you for the pertinent suggestion. We have included "Oral nutritional supplementation" as a keyword, and its the definition and indication of use in the introduction section, in the fifth paragraph.

Authors’ answer 4: We apologize for the mistake and thank you for the revision. We have included the citations of these references throughout the text, in the following places:

- Reference 5 = in the second paragraph of Introduction.

- References 30 and 31 = in the first paragraph of Materials and Methods, item 2.1 (Study design and participants).

We also inform you that we conducted a new search on recently published articles related to the topic. As a result, we have included new references that were cited in the introduction and discussion. 

Reviewer 2 Report

The manuscript written is of good quality. The statistical methods used for the data analysis are correct.

Just a one minor note: the references 4th, 31st, 32nd, 34th, 59th and 42nd points to the article written in Portuguese. If to write the title not in English is in accordance with the journal’s requirements – that’s fine, if not – that should be translated into English with a mark [In Portuguese].

And possible corrections:

Line 129: I would suggest changing “…in clinical trials with people with chronic injuries…” to “in clinical trials in people with chronic injuries”

Line 134: I would suggest changing “…according to protocols described in other clinical trials for analysis…” to “…according to the protocols described in other clinical trials for the analysis…”

I will not correct other “the” – there may be some more.

Line 129: I would suggest changing “…but two withdrew from…” to “…but two were withdrawn from…”

Line 159: A point in “…saturated fatty. acids” should be deleted.  

Line 280: in the phrase “…venous ulcers are have been rare…” are should probably be deleted. 

Author Response

Authors' response 1: Thank you for your suggestion. We have modified the titles for their English versions, although the journal's guidelines do not require articles in languages other than English to have their titles translated.

Authors’ answer 2: We are thankful for the corrections you pointed out. We made all the suggested changes and inform you that we have requested a new thorough reading by a native English-speaking proofreader.